# Hemoglobin and Its Relationship with Fatigue in Long-COVID Patients Three to Six Months after SARS-CoV-2 Infection

**DOI:** 10.3390/biomedicines12061234

**Published:** 2024-06-01

**Authors:** Somayeh Bazdar, Lizan D. Bloemsma, Nadia Baalbaki, Jelle M. Blankestijn, Merel E. B. Cornelissen, Rosanne J. H. C. G. Beijers, Brigitte M. Sondermeijer, Yolanda van Wijck, George S. Downward, Anke H. Maitland-van der Zee

**Affiliations:** 1Department of Pulmonary Medicine, Amsterdam UMC, University of Amsterdam, 1105 AZ Amsterdam, The Netherlands; s.bazdar@amsterdamumc.nl (S.B.); l.d.bloemsma@amsterdamumc.nl (L.D.B.); j.m.blankestijn@amsterdamumc.nl (J.M.B.); m.e.b.cornelissen@amsterdamumc.nl (M.E.B.C.);; 2Amsterdam Institute for Infection and Immunity, Amsterdam UMC, 1105 AZ Amsterdam, The Netherlands; 3Amsterdam Public Health Research Institute, Amsterdam UMC, 1105 AZ Amsterdam, The Netherlands; 4Department of Respiratory Medicine, Nutrim Institute of Nutrition and Translational Research in Metabolism, Faculty of Health, Medicine and Life Sciences, Maastricht University, 6202 AZ Maastricht, The Netherlands; r.beijers@maastrichtuniversity.nl; 5Department of Pulmonology, Spaarne Gasthuis, 2035 RC Haarlem, The Netherlands; bsondermeijer@spaarnegasthuis.nl; 6Department of Environmental Epidemiology, Institute for Risk Assessment Sciences (IRAS), Utrecht University, 3584 CL Utrecht, The Netherlands; g.s.downward@uu.nl; 7Department of Global Public Health & Bioethics, Julius Center for Health Sciences and Primary Care, University Medical Center Utrecht, 3584 CX Utrecht, The Netherlands

**Keywords:** Long-COVID, anemia, hemoglobin, inflammation, fatigue

## Abstract

*Background:* While some long-term effects of COVID-19 are respiratory in nature, a non-respiratory effect gaining attention has been a decline in hemoglobin, potentially mediated by inflammatory processes. In this study, we examined the correlations between hemoglobin levels and inflammatory biomarkers and evaluated the association between hemoglobin and fatigue in a cohort of Long-COVID patients. *Methods:* This prospective cohort study in the Netherlands evaluated 95 (mostly hospitalized) patients, aged 40–65 years, 3–6 months post SARS-CoV-2 infection, examining their venous hemoglobin concentration, anemia (hemoglobin < 7.5 mmol/L in women and <8.5 mmol/L in men), inflammatory blood biomarkers, average FSS (Fatigue Severity Score), demographics, and clinical features. Follow-up hemoglobin was compared against hemoglobin during acute infection. Spearman correlation was used for assessing the relationship between hemoglobin concentrations and inflammatory biomarkers, and the association between hemoglobin and fatigue was examined using logistic regression. *Results:* In total, 11 (16.4%) participants were suffering from anemia 3–6 months after SARS-CoV-2 infection. The mean hemoglobin value increased by 0.3 mmol/L 3–6 months after infection compared to the hemoglobin during the acute phase (*p*-value = 0.003). Whilst logistic regression showed that a 1 mmol/L greater increase in hemoglobin is related to a decrease in experiencing fatigue in Long-COVID patients (adjusted OR 0.38 [95%CI 0.13–1.09]), we observed no correlations between hemoglobin and any of the inflammatory biomarkers examined. *Conclusion:* Our results indicate that hemoglobin impairment might play a role in developing Long-COVID fatigue. Further investigation is necessary to identify the precise mechanism causing hemoglobin alteration in these patients.

## 1. Introduction

Over four years ago, the World Health Organization (WHO) declared COVID-19 a global pandemic. However, infections from SARS-CoV-2 and their long-term consequences remain a global challenge [1,2]. Of particular concern are the many aspects of the long-term outcomes that remain unknown, especially among those who have survived a SARS-CoV-2 infection [3,4]. The term “Long-COVID” refers to the condition in which patients experience a variety of persisting symptoms long after the acute phase of the infection has resolved but often typified by fatigue, dyspnea, brain fog, sleep problems, muscle stiffness, and mental problems [1,5,6]. These symptoms are not restricted to pulmonary manifestations; and just like the acute phase of the disease, organs other than the lungs may be impacted [7,8]. However, many long-term symptoms may not have been present in the acute phase.

Long-COVID is causing widespread disability worldwide, with many potentially developing lifelong disabilities. Nevertheless, current diagnostics and treatments are insufficient, necessitating urgent clinical investigations to address hypothesized underlying biological mechanisms [1]. Whilst the mechanism of long-COVID continues to be unclear due to the wide clinical spectrum and organ involvement, only a thorough understanding of its pathophysiology will allow us to assess and treat the disease’s side effects [9]. While persistence of respiratory symptoms, particularly dyspnea and cough, appear to be common in Long-COVID, some of the other persistent symptoms of Long-COVID may be part of a multisystem disease, and interdisciplinary treatment of these chronic sequelae is necessary [10].

A decrease in hemoglobin may be one of the non-respiratory complications of a SARS-CoV-2 infection. In the acute phase of infection, a reduction in hemoglobin is likely mediated via interactions between viral agents and host cells and has previously been found to be a predictive factor for severe respiratory failure in pneumonic COVID-19 patients [11,12]. Based on the results of multivariate regression of a study comprising 1147 COVID-19 patients across five tertiary pediatric hospitals in Asia, higher levels of hemoglobin and platelets were protective against severe/critical COVID-19 infection [11]. A retrospective investigation of 23 COVID-19 hospitalized patients with pneumonia also revealed a significant association between higher levels of hemoglobin and a reduced need for mechanical ventilation administration, with an odds ratio of 0.313 [12]. Various mechanisms for decreased hemoglobin have been proposed, including the conversion of Hb into other forms such as methemoglobin (MetHb) and carboxyhemoglobin (COHb) [13]. While oxidative stress, which is linked to infectious diseases, may contribute [14], medications given during treatment may also be relevant, given that certain medicines such as hydroxychloroquine used to treat COVID-19 have a high potential for oxidation, which can lead to MetHb and COHb [15]. An increase in COHb during SARS-CoV-2 infection may also potentially be related to hemolytic anemia, and an increase in MetHb is a result of hemoglobin oxidation that could also be driven by nitric oxide (NO) [13]. Besides oxidative stress, hyperinflammation may be another reason for developing new-onset anemia during SARS-CoV-2 infection [16]. Investigations of hospitalized patients with COVID-19 disease revealed that increased inflammatory indices such as Interleukin 6 (IL-6) and C-reactive protein (CRP) were increased in those with COVID-19-induced anemia [17].

Anemia is defined as a decrease in hemoglobin (Hb) concentrations and/or red blood cell (RBC) absolute counts leading to an inadequate supply of physiological requirements, and its common diagnostic method is to assess the Hb level [18]. This hematologic disease is typically a multifaceted medical condition, and different factors such as the patient’s medical history and the underlying pathological mechanism(s) should be considered while diagnosing and categorizing it [19]. Common manifestations of anemia include a wide range of symptoms such as pale skin, shortness of breath, palpitations, chest pain, dizziness, loss of strength, weakness, and fatigue [20]. According to studies on the prevalence of fatigue and its associated factors, anemia is one of the most frequently found risk factors linked to fatigue in patients with chronic kidney disease (CKD) [21].

Fatigue is the most common symptom in Long-COVID patients [22]. While anemia, psychosocial problems, sleep disorders, and sleep-related respiratory disorders are considered common causes of fatigue in general, cancer, thyroid dysfunction, and other somatic diseases can also be reasons [23]. In a study of 42 men and 33 women, Pasini et.al. reported a mean hemoglobin of 11.5 g/dL (normal range: 14–18 g/dL) among men and 10.9 g/dL (normal range: 12–16 g/dL) among women 60 days following their hospital discharge after COVID-19 [24]. Furthermore, in a 6-month follow-up of 412 ex-COVID-19 patients, it was reported that the average hemoglobin level was significantly lower in patients suffering from fatigue compared to those who were not (13.8 vs. 14.3 g/dL). However, after performing multivariate regression, no statistically significant associations were observed between hemoglobin, other cell blood count measurements (including inflammatory biomarkers), and Long-COVID symptoms [25].

A better understanding of the mechanisms involved in Long-COVID and hemoglobin status/anemia will allow better treatment of those suffering from this condition. While there is some evidence that Long-COVID patients have decreased hemoglobin levels, its link with inflammatory indices as well as its role in developing fatigue symptoms remain unclear. Finding biomarkers that can help to direct optimized care to patients most at risk is crucial, and since hemoglobin is easily assessed by assay and has been associated with Long-COVID, we believe that it is a biomarker worthy of further investigation. The purpose of the current study is therefore to examine the correlations between hemoglobin levels and inflammatory biomarkers and to evaluate the potential role of the inflammatory process in decreasing hemoglobin levels in Long-COVID patients. Furthermore, this study investigates the association between hemoglobin and fatigue in a cohort of Long-COVID patients to evaluate whether hemoglobin levels contribute to developing fatigue in these patients.

## 2. Methods and Materials

### 2.1. Study Setting and Participants

The population for the current study consists of participants of the Precision Medicine for More Oxygen (P4O2) COVID-19 cohort. Details on the study design of this cohort have been published elsewhere [26]. In brief, P4O2 COVID-19 is a prospective cohort study that recruited 95 Long-COVID patients, aged 40–65 years, from outpatient Long-COVID clinics in five hospitals in the Netherlands. Two research visits were performed for each patient between months 3–6 and months 12–18 after SARS-CoV-2 infection, during which questionnaires were administered, and physical examinations and biological sampling were performed. There were no specific exclusion criteria for the FU study visit of this cohort, and nine months following the initial study visit, all participants received invitations for a follow-up appointment, but a portion was lost to follow-up. Information about the acute phase of the disease was extracted from patients’ medical charts. Participants for whom a hemoglobin measurement was available, either during *SARS-CoV-2* infection or their first follow-up (FU) visit 3–6 months after SARS-CoV-2 infection, or both, were eligible for analysis in the current paper. The second study visit was conducted 12–18 months after infection, which included a fairly high number of lost to FU among whom only 11 provided blood samples for hemoglobin.

### 2.2. Ethical Considerations

The study protocol for the P4O2 COVID-19 cohort was approved by the ethical committee of Amsterdam UMC (reference number NL74701.018.20), and all participants provided written informed consent prior to their enrolment.

### 2.3. Variables in This Study

Measurements during the acute phase of the SARS-CoV-2 infection (baseline) and FU laboratory measurements of hemoglobin were assessed via venipuncture, performed under aseptic conditions, and analyzed following standard laboratory procedures in each participating hospital. To be eligible for the current study, a hemoglobin level either at baseline or FU (n = 93) was required. A subset of patients (n = 27) did not provide standard venipuncture at the FU visit but most of them did provide a hemoglobin value measured via finger prick test during pulmonary function testing (generally occurring concurrently with the FU visit), which was used instead, and this way all 95 participants were eligible for the current analysis. As this method was expected to provide results somewhat different from those of venipuncture, the hemoglobin values from these participants were only used to examine whether their hemoglobin levels changed by >10% (i.e., clinically significant difference, as described below). Anemia was defined, as per the reference range at the measuring laboratory, as a hemoglobin level less than 7.5 mmol/L in women and less than 8.5 mmol/L in men according to the reference laboratory range. Absolute changes in hemoglobin were calculated as the difference between FU and baseline hemoglobin and were classified into increased, decreased, or unchanged hemoglobin. In addition, “clinically relevant” changes in hemoglobin were determined by an increase or decrease in hemoglobin of 10% or more. As there is no universal definition of a clinically significant change in hemoglobin levels, we utilized the determination of obstetrics and gynecology references which usually recognize a decrease in hemoglobin level if it is more than 10% decrease [27,28,29].

The Fatigue Severity Scale (FSS) standard questionnaire was used to categorize patients into those with or without fatigue. The total FSS was the sum of the nine questions and the average FSS was calculated by dividing this sum by nine [30]. An average score equal to or more than four was considered as a positive indicator of fatigue, similar to previous research in the P4O2 COVID study [26]. The fatigue severity scale originally utilized a cut-off of 4 or more to indicate severe fatigue. We acknowledge that different cut-offs have been developed, including low (<4), medium/borderline (4–5), and high/severe (>5) [31]. However, for simplicity, and consistency with our other work in this population [26] we have chosen to consistently use 4 as a cut-off point.

Blood samples were also examined for a variety of biomarkers, and markers that indicated the presence of an inflammatory process were included as inflammatory indices in the current study. Identified markers were D-Dimer, Neutrophil/Lymphocyte ratio (NLR), Immature Granulocyte (IG), C-reactive protein (CRP), Creatine phosphokinase (CPK), and Lactate Dehydrogenase (LDH) [32,33,34]. Normal ranges and analytical approaches were as per the standard operating procedures at participating hospitals.

Additional variables assessed in this paper were: general demographics (i.e., age, sex, and ethnicity), past medical history of underlying diseases (e.g., hypertension, diabetes, cardiovascular disease, kidney diseases, asthma, etc.), history of hospital stay (including length and any ICU admission), dominant SARS-CoV-2 variant during admission, body mass index (BMI), vaccination status, and glomerular filtration rate (GFR—based on serum creatinine).

### 2.4. Statistical Analysis

Continuous variables were described by mean (SD) for variables with a normal distribution (age, BMI, baseline hemoglobin, FU hemoglobin, hemoglobin change) or median (25th percentile, 75th percentile) for those that were not. Categorical data were described by frequency (percent). Characteristics were also stratified by sex since the reference range for anemia differs between males and females. The Paired-Samples T-test was utilized to compare parametric variables between two timepoints (baseline and first FU visit) and the Independent Samples Test was used to compare parametric variables between two different groups (fatigue patients vs. non-fatigue patients, male vs. female). Chi-square testing was employed while comparing categorical variables between these groups, and for comparing non-parametric variables between them, the Mann–Whitney U test was used. The correlations between hemoglobin levels and inflammatory biomarkers were evaluated via Pearson’s correlation test for parametric values and Spearman’s correlation for non-parametric ones. A false discovery rate (FDR) was not used as it can be overly conservative especially when validating a smaller number of previously described features or in cases where the effective number of tests is smaller (due to redundancy in similar features measuring the same latent variables) [35,36]. Finally, logistic regression models were developed to assess the relationships between fatigue at 3–6 months after infection and hemoglobin levels (1) in the acute phase, (2) 3–6 months after SARS-CoV-2 infection, and (3) the difference in hemoglobin levels between infection and follow-up. Fatigue was defined as a binary (yes/no) variable based on the pre-defined FSS cutoff of equal or greater than four (yes) or less than four (no). Unadjusted models and models adjusted for age and sex were generated. Covariates for adjustment were identified through the generation of a Directed Acyclic Graph (DAG) (Figure 1) [37]. According to this DAG, age, sex, ethnicity, BMI, underlying diseases, hospital admission, ICU admission, inflammation, and lung function were all identified confounders for the potential pathway of developing fatigue due to hemoglobin alteration. However, considering the sample size and a target of 20 events per covariate, only age and sex, identified as the most important potential confounders, were included in the regression analysis.

Since renal function has the potential to affect both hemoglobin and fatigue, we performed a sensitivity analysis by restricting the regression analysis to those with GFR at follow-up greater than 60 mL/min/1.73 m^2^ (n = 56). This cut-off was chosen to differentiate those with normal/mild kidney dysfunction from more severe cases where the prevalence of anemia is higher [38,39,40]. Furthermore, since a case was detected among the participants with an extreme outlier value for hemoglobin (both baseline and FU), another sensitivity analysis was also performed after excluding this single case. All statistical analyses were performed by SPSS version 28 and a *p*-value ≤ 0.05 was considered statistically significant for all statistical tests.

## 3. Results

In total, 1 or more hemoglobin results were available from all 95 participants in the P4O2 COVID cohort, with 59 having data on the Hb level at both the acute phase and at FU (11 participants had missing data during the acute phase and 27 at FU). The mean age of our study population was 54.2 years, and 50.5% of them were men. Most of the participants (89.5%) were suffering from some form of underlying disease, and comorbidities were more common in men than in women (93.8% vs. 85.1%). According to the WHO severity index, most of the infections were categorized as moderate (61.3%) or severe (28.0%). The characteristics that were significantly different between men and women were fatigue average score during the Long-COVID phase (83.3% in females and 68.9% in males, *p*-value = 0.026 according to Independent Samples Test), ICU admission (19.1% in women and 38.3% in men, *p*-value = 0.044 according to Chi-Square test), ICU stay (median 0 (0–0) days in women and median 0 (0–8) days in men, *p*-value = 0.044 according to Mann–Whitney U test), and BMI (32.2 in women and 28.9 in men, *p*-value = 0.002 according to Independent Samples Test) (Table 1).

### 3.1. Hemoglobin Levels

The mean hemoglobin level during SARS-CoV-2 infection for our entire study population was 8.6 mmol/L (8.35 mmol/L in women and 8.84 mmol/L in men) (Table 2), which is within the normal laboratory reference range (>7.5 in women and >8.5 in males). During the acute phase, the average hemoglobin concentration was 8.61 mmol/L and 8.70 mmol/L 3–6 months after infection (*p*-value for difference: 0.003 according to Paired-Samples T-test). Hemoglobin values were significantly different (via Independent Samples Test) between men and women both during the acute phase (8.84 vs. 8.35 mmol/L, respectively, *p*-value = 0.023) and at follow-up (9.04 vs. 8.36 mmol/L *p*-value = 0.003,) (Table 2). In patients with hemoglobin levels available at both time points (n = 59), the hemoglobin levels increased on average by 0.3 (from 8.41 mmol/L during the acute phase and 8.71 mmol/L 3–6 months after infection). This was more pronounced in men where the average hemoglobin increased from 8.89 mmol/L to 9.24 mmol/L (mean difference = 0.42 mmol/L). For female participants, the mean hemoglobin value remained the same at baseline and FU (8.35 mmol/L and 8.36 mmol/L, respectively, mean difference = 0.18 mmol/L). However, men and women’s differences in hemoglobin change did not reach statistical significance (*p*-value = 0.222 according to Independent Samples Test). Consistent with this overall increase, hemoglobin levels at FU were higher for 66.1% of our participants (N = 39), with 14 having an increase of greater than 10%. However, 28.8% (N = 17) of the participants had lower hemoglobin levels at FU, with 8 showing a decrease of greater than 10%. Also, in three cases the hemoglobin level during FU remained the same as the baseline value.

The prevalence of anemia during the acute phase of SARS-CoV-2 infection was 22.6% (19 cases, among whom 7 continued experiencing anemia during follow-up, 8 did not, and data were unavailable for four). This reduced to 16.4% at follow-up (n = 11, 3 were new-onset and 7 were persistent from the acute phase, and data were missing for 1, Appendix A). Suffering from anemia was more common in men compared to women during both phases of COVID-19 disease (31.1% versus 12.8% during the acute phase (*p*-value = 0.046 according to Chi-Square Test) and 18.8% versus 13.9% during the Long-COVID phase (*p*-value = 0.587 according to Chi-Square Test).

We also examined demographic and clinical characteristics, by dividing the population into notable hemoglobin change (notable decrease versus notable increase or no change). Of note, the group with a notable decrease in hemoglobin had a longer hospital stay than those without (17 days vs. 8 days, Appendix A).

When comparing hemoglobin levels between those with and without fatigue 3–6 months after infection, we observed that those experiencing fatigue had lower levels of hemoglobin both at baseline (8.53 vs. 8.89 mmol/L, respectively, *p*-value = 0.171 according to Independent Samples Test) and at FU (8.54 vs. 9.24 mmol/L, *p*-value = 0.046 according to Independent Samples Test) than those without fatigue (Table 2). Boxplots displaying the difference in hemoglobin level between those with fatigue versus those without are shown in Figure 2.

### 3.2. Correlations between Hemoglobin and Inflammatory Biomarkers

Results of Spearman correlation tests demonstrated that there were no strong or statistically significant correlations between hemoglobin levels (at baseline, follow-up, and the change in hemoglobin levels) and inflammatory biomarkers (Table 3). Among other variables that were found to be linked with hemoglobin level according to the DAG (Figure 1), the duration of hospital stay, as well as ICU stay had a positive correlation with hemoglobin change (rs = 0.268, and rs = 0.360, respectively) (Appendix A).

### 3.3. Associations between Hemoglobin Levels and Fatigue

Results of logistic regression analysis indicated that higher venous hemoglobin concentrations were associated with lower odds of fatigue 3–6 months after infection (Table 4). While findings were non-significant, an association between increased Hb, and reduced fatigue was consistently observed. The strongest associations were observed with hemoglobin levels 3–6 months after infection where increased hemoglobin was associated with a reduced likelihood of fatigue. A 1 mmol/L increase in hemoglobin was associated with an adjusted odds ratio of 0.35 for fatigue (95%CI 0.12–1.02). Another strong relationship between fatigue and hemoglobin indices was observed for the change in hemoglobin levels, where a 1 mmol/L positive increase in Hb (i.e., greater in visit 2 than visit 1) resulted in an adjusted OR of 0.38 (95% CI 0.13–1.09). There was also a positive association between fatigue and anemia at both baseline and at follow-up (adjusted OR 1.56 [95% CI 0.42–5.78] and OR 2.25 [95% CI 0.25–20.20], respectively).

Results of the sensitivity analysis where patients with impaired renal function were excluded showed associations that were directionally consistent and comparable to those in the total population. For example, the adjusted ORs of the associations between fatigue and FU hemoglobin and hemoglobin change were, respectively, 0.475 (95% CI 0.159–1.419) per 1 mmol/L increase in hemoglobin, and 0.467 (95% CI 0.150–1.452) per 1 mmol/L increase in hemoglobin (Appendix A). In addition, the results of the other sensitivity analysis, which excluded the case with the extreme outlier data for the hemoglobin variable, showed barely any variation from the findings of the regression analysis in the total population (Appendix A).

## 4. Discussion

In the current study, we aimed to assess the role of hemoglobin levels during the acute phase of SARS-CoV-2 infection and 3–6 months after follow-up against inflammatory biomarkers and fatigue in the P4O2 COVID-19 study. Overall, we found that hemoglobin levels increased between the acute and follow-up phases and that the average hemoglobin concentrations in our study population were within the normal range. The proportion of anemic patients also decreased, from 22.6% to 16.4%. However, hemoglobin levels were significantly lower, both at baseline and at follow-up, in patients with fatigue compared to patients without fatigue. No correlations were detected between hemoglobin and inflammatory biomarkers.

Other work examining anemic COVID-19 patients has reported that the SARS-CoV-2 virus is a contributor to both causing and exacerbating anemia [39]. Further, anemia itself has also been shown to play a role in prolonging COVID-19 symptoms [41]. It is therefore possible that anemia, either pre-existing or caused by acute infection, may contribute to some (but not all) of the fatigue symptoms in those who suffer from Long-COVID [42]. In a similar study of 145 former COVID-19 patients, it was reported that while the prevalence of anemia, hyperferritinemia, and systemic thrombo-inflammation all gradually decreased, a sizable portion of the post-COVID population still showed persistent anemia (4.65–11.1%) or iron deficiency (16–35%) at different FU evaluation time points, which may have contributed to their ongoing symptom burden, including fatigue [43].

Fatigue is a multifaceted condition and although its specific etiology can be uncertain, it is usually identified as the main manifestation of anemic patients [44,45] Our regression analysis demonstrated that hemoglobin levels are lower in Long-COVID patients who suffer from fatigue in both the acute phase and 3–6 months afterward. This suggests that hemoglobin may play a role in the fatigue symptoms of Long-COVID patients. Nevertheless, as the current study is observational in nature, further investigation is necessary to establish causation as the link between anemia and fatigue is not always evident. For example, a review of the contributors to persistent fatigue in Myelodysplastic syndrome patients revealed that no clear association has been found between fatigue and anemia. [46]. Therefore, other potential factors including inflammation, sleep-related problems, and oxidative stress may provide alternate explanations.

Another finding in our study suggesting a role of hemoglobin in fatigue is that we also observed a higher percentage of women who reported fatigue (83.3% in women vs. 68.9% in men), while their mean hemoglobin level in the Long-COVID phase was lower than in men (8.36 vs. 9.04). However, BMI was also significantly different between men and women, which can also be associated with fatigue [47,48]. Another characteristic that varied between males and females was hemoglobin change, where men showed a higher rate of increase than women. Previous research has reported that increased erythropoiesis is a secondary outcome of higher testosterone concentrations in men [49], which may explain the difference observed here.

These findings may have potential clinical implications, providing evidence for physicians to consider monitoring the hemoglobin levels in their Long-COVID patients, particularly in those suffering from fatigue as their primary symptom. Whether fatigue can be managed by treating any underlying hemoglobin impairment is outside the scope of the current work; however, it is worth closer evaluation.

We did not observe any correlation between inflammatory markers and hemoglobin concentrations. This stands in contrast to other work such as that of Lanser et al., who examined inflammation-mediated anemia, reporting the impact of *immune* system stimulation and cytokine-mediated alterations on iron homeostasis [16]. The three inflammatory variables that they assessed in their study were ferritin, IL-6, and CRP—of which we only studied CRP. This difference may illustrate that only extremely severe inflammation can affect hemoglobin values, that our sample size was insufficient to identify an effect, or that the inflammatory biomarkers available for use in the current study were not optimal for the evaluation of inflammatory-driven Hb disruption. For example, the erythropoiesis-suppressing effects of specific inflammatory cytokines, such as TNFα, IL-1, and interferon-γ, have been reported in various investigations. Therefore, a broader assessment of inflammatory markers alongside other diagnostic tests for anemia related to inflammation (serum iron, ferritin, and hepcidin), may be a more accurate way to assess the correlation between anemia and inflammation [50]. For example, in an analysis of 214 ex-COVID-19 patients, where all aforementioned variables were included, patients who reported Long-COVID Syndrome months after the acute phase were distinguished by inflammation and abnormal iron homeostasis that lasted longer than two weeks after the onset of symptoms. This led the researchers to conclude that the effects of inflammatory iron dysregulation on erythropoiesis and blood oxygen transportation may be partially responsible for several of the characteristics of Long-COVID [51]. Alternately this might suggest that factors other than inflammation have a more significant role. For example, there is evidence that SARS-CoV-2 infection can produce hemolytic anemia by either generating structural changes in host cells (RBCs) via viral invasion or activating autoantibodies [52]. The fact that we could not validate the same link between inflammation and Hb that was reported by Hanson et al. emphasizes the need to further test the robustness of biomarker analyses [51]. Furthermore, a meta-analysis reviewing the studies evaluating the COVID-19 biomarkers revealed a number of findings, including different results per population, which highlights the significance of validation studies [53]. Especially, since long COVID is a relatively young disease correlations with biomarkers in one population should be validated in other populations. Additional work is therefore needed to further understand the mechanism of hemoglobin decrease in SARS-CoV-2 infection, and how (or if) these factors persist long-term.

An alternative to the proposed hemoglobin and inflammatory biomarkers explored here is the evidence-based association with oxidative stress [54]. Experiments have shown that oxidative stress significantly impacts the structure of hemoporphyrin, which in turn affects hemoglobin function and reduces the efficiency with which hemoglobin binds to oxygen. This process is caused by lipid peroxidation (LPO), which is followed by the formation of LPO’s primary and end products [55]. Understanding the potential role of oxidative stress in the development of hemoglobin decrease would be a significant finding that may lead to preventative and therapeutic approaches [56,57].

Among this study’s strengths is its well-defined population of Long-COVID patients, who have comprehensive information, including hospital and personal data. Additionally, these patients have an extensive list of fatigue symptoms obtained through standard fatigue questionnaires. However, this study also has some limitations. First, while SARS-CoV-2 infection is the exclusive focus of this investigation, it has been revealed that other viral infections, such as seasonal influenza, may have long-term effects similar to those following SARS-CoV-2 infection [58]. Thus, it is uncertain whether our findings are specific to this particular viral infectious disease, or indicative of post-viral syndromes more broadly. Our limited sample size restricts the overall statistical power, including for regression analysis, and limits the external generalizability of our findings. Further compounding this was missing values for several variables of interest, particularly the baseline variables of inflammatory biomarkers. Such data were only available if they had been ordered during the original presentation and admission in different hospitals. Imputation was considered but ultimately decided against as these were outcome variables. Towards the later part of the pandemic, follow-up visits were more often occurring remotely, meaning that biological sampling at this point was reduced. Furthermore, not having pre-infection hemoglobin data makes evaluating acute changes impossible. In addition, it has been shown that vaccination against SARS-CoV-2 might enhance the likelihood of hematologic problems, such as anemia [59] and on the other hand, that receiving the vaccine prior to SARS-CoV-2 infection might be associated with a lower risk of developing Long-COVID [60]. However, since the vaccination pattern varied greatly (Appendix A) and our sample size was not large enough, we were unable to analyze the association between anemia and vaccination status in our population. Finally, this cohort was started at a point in time when not much was known about Long-COVID and therefore no specific age group was known to be important at that time, and it seemed that it was mainly affecting individuals of 40 years and older. In addition, because older patients (above 65) are more likely to have more comorbidities that may interfere with the analysis we decided to limit our population to below that age. Thus, the age range of our study population (40 to 65 years) makes us more cautious in generalizing our findings to other age ranges.

## 5. Conclusions

In our study of Long-COVID patients, we observed that those with fatigue tended to have lower hemoglobin levels than those without. These findings suggest a role of hemoglobin in Long-COVID fatigue. To determine the causes of low hemoglobin levels in Long-COVID patients and in turn, the best course of action for treatment, further studies on the mechanisms influencing hemoglobin levels in Long-COVID are necessary. Furthermore, it is important to perform validation studies to examine whether our findings are reproducible and robust across different cohorts.

## Figures and Tables

**Figure 1 biomedicines-12-01234-f001:**
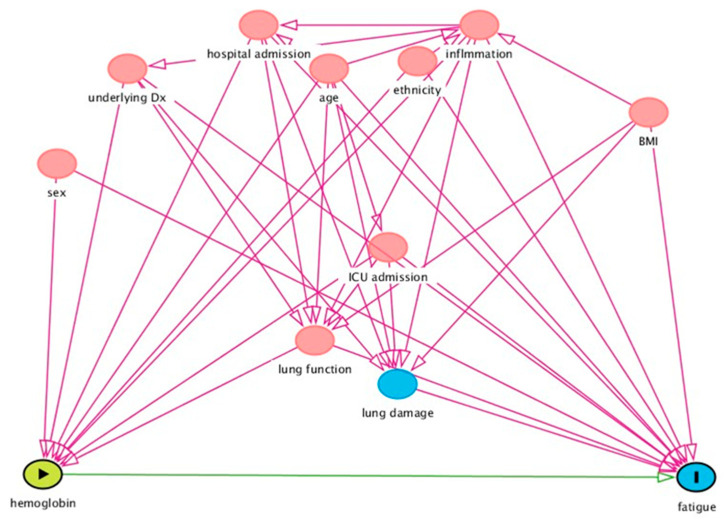
DAG for determining the confounders and effect modifiers in the relationship between hemoglobin levels and fatigue. Figure legend; 
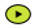
 exposure, 
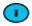
 outcome, 
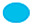
 ancestor of outcome, 
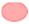
 ancestor of exposure and outcome, 
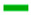
 (potentially) causal path, 
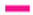
 biasing path.

**Figure 2 biomedicines-12-01234-f002:**
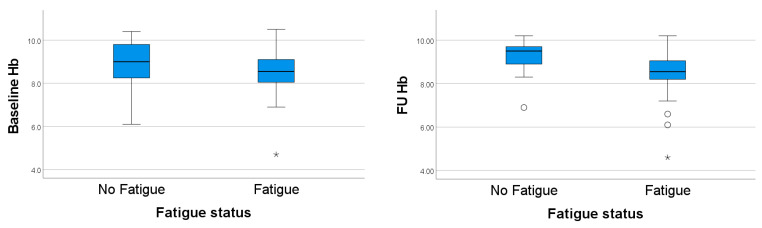
Boxplots for assessing the difference in hemoglobin levels between fatigue and non-fatigue cases. Figure legend: O: mild outlier, *: extreme outlier. *Footnotes:* 1. fatigue status data were missing for 8 cases, and considering the missing data for Hb variables, this graph includes information from 52 cases. 2. The extreme outlier in both baseline and FU figure belongs to the same case.

**Table 1 biomedicines-12-01234-t001:** Characteristics of the study population.

Population Characteristics			Total (N = 95)	*p*-Value £	Female (N = 47)	Male (N = 48)
	**Demographic characteristics**					
Age (mean, SD)			54.2 (6.2)	0.765	54.3 (6.1)	54.0 (6.3)
Ethnicity (N, %)		African	8 (9.4%)	0.077	5 (1.6%)	3 (7.0%)
		Asian	3 (3.5%)		2 (4.7%)	1 (2.3%)
		Caucasian	66 (76.7%)		33 (76.7%)	33 (76.7%)
		Latin-American	3 (3.5%)		3 (7.0%)	0 (0.0%)
		Others	6 (7.0%)		0 (0.0%)	6 (14.0%)
	**Clinical characteristics**					
Suffering from any comorbidities (N, %)			85 (89.5%)	0.170	40 (85.1%)	45 (93.8%)
BMI (mean, SD)			30.39 (5.3)	0.002	32.15 (5.4%)	28.85 (4.7)
Dominant SARS-CoV-2 variant at the time of infection (N, %)		Alpha	43 (45.3%)	0.278	23 (48.9%)	20 (41.7%)
		Delta	41 (43.2%)		21 (44.7%)	20 (41.7%)
		Omicron	11 (11.6%)		3 (6.4%)	8 (16.7%)
Hospitalized (N, %)			85 (89.5%)	0.170	40 (85.1%)	45 (93.8%)
Hospital stay (days) (median, percentile 25–75) *			8.00 (4.0–15.2)	0.083	8.00 (3.0–13.0)	10.00 (5.0–21.0)
ICU admission (N, %)			27 (28.7%)	0.40	9 (19.1%)	18 (38.3%)
ICU stay (days) (median, percentile 25–75) *			0.00 (0.0–4.0)	0.044	0.00 (0.0–0.0)	0.00 (0.0–8.0)
One dosage of SARS-CoV-2 vaccination			66 (69.5%)	0.771	32 (68.1%)	34 (70.8%)
Two dosages of SARS-CoV-2 vaccination			41 (43.2%)	0.261	23 (48.9%)	18 (37.5%)
WHO severity index (N, %)		Mild	10 (10.8%)	0.240	7 (15.2%)	3 (6.4%)
		Moderate	57 (61.3%)		30 (65.2%)	27 (57.4%)
		Severe	26 (28.0%)		9 (19.6%)	17 (36.2%)
Average FSS during the 3–6 months visit (median, percentile 25–75) ¥			5.6 (4.1–6.3)	0.026	5.8 (4.7–6.4)	5.3 (3.0–6.3)
Fatigue (Average FSS ≥ 4) during the 3–6 months visit (N, %) ¥				0.116		
Yes			66 (69.5%)		35 (74.5%)	31 (68.9%)
No			21 (22.1%)		7 (14.9%)	14 (29.2%)
Missing			8 (8.4%)		5 (10.6%)	3 (63%)

* The number of days spent in the ICU or hospital for the participants who were not admitted were counted as 0. ¥ Fatigue data were missing for eight (five female and three male) participants. £ *p*-value for comparing variables between men and women.

**Table 2 biomedicines-12-01234-t002:** Hemoglobin level status and the prevalence of anemia in the P4O2 COVID-19 study.

Hb and Anemia Status		Total	Female	Male	*p*-Value	Experiencing Fatigue 3–6 Months after Infection ¶	No Fatigue 3–6 Months after Infection	*p*-Value π
Total number of participants		95	47	48	---	66	21	---
Hb ◊ level during the acute phase (mmol/L) (mean, SD) (n = 84)		8.61 (0.99)	8.35 (0.86)	8.84 (1.05)	**0.023**	8.53 (0.99)	8.89 (1.06)	0.171
Hb level during the 3–6 months visit (mmol/L) (mean, SD) (n = 68)		8.70 (0.98)	8.36 (0.87)	9.04 (1.00)	**0.004**	8.54 (0.99)	9.24 (0.99)	**0.046**
Hb level change (mmol/L) (mean, SD) ¥ (n = 59)		0.30 (0.74)	0.18 (0.80)	0.42 (0.67)	0.222	0.19 (0.73)	0.68 (0.69)	0.069
Clinically notable (10%) Hb change (N, %)	Hb decrease ‡	8 (8.4%)	4 (10.50%)	4 (9.50%)	0.881	6 (10.90%)	1 (5.90%)	1.000
	Hb increase Ⱡ	15 (23.7%)	6 (15.80%)	9 (21.40%)	0.519	10 (18.20%)	3 (17.6%)	1.000
Hb level change µ (N, %)	Hb decrease *	17 (28.8%)	10 (34.50%)	7 (23.30%)	0.344	14 (32.20%)	1 (10.00%)	0.249
	Hb increase €	39 (66.1%)	19 (65.50%)	20 (66.70%)	0.926	26 (60.50%)	9 (90.00%)	0.137
Anemia during the acute phase (N, %) £ (n = 84)		19 (22.61%)	5 (12.80%)	14 (31.10%)	**0.046**	13 (23.2%)	4 (20%)	1.000
Anemia during the 3–6 months visit (N, %) £ (n = 68)		11 (16.41%)	5 (13.90%)	6 (18.80%)	0.587	10 (19.2%)	1 (10%)	0.674

*Footnotes:* ¶ fatigue data were missing for eight cases. π the *p*-values are related to the comparisons that have been performed for Hb levels/anemia status, between patients experiencing fatigue 3–6 months after infection and those who did not. ◊ normal Hb range is >7.5 in women and >8.5 in males. ¥ follow-up Hb–baseline Hb (difference between baseline and follow-up Hb; there were missing data for 36 (37.90%) cases. ‡ cases with follow-up Hb–baseline Hb < 0.10 × baseline Hb. Ⱡ cases with baseline Hb–follow-up Hb < 0.10 × baseline Hb. * cases with follow-up Hb < baseline Hb. € cases with follow-up Hb > baseline Hb. £ Hb level less than 7.5 mmol/L in women and less than 8.5 mmol/L in men. µ For three cases the baseline Hb was exactly the same as the FH Hb and they had no Hb change. *p*-values showing statistically significant difference between compared groups have been marked in bold font.

**Table 3 biomedicines-12-01234-t003:** Correlations between hemoglobin measures and inflammatory indices.

Variable	Median (Percentile 25–75)	cc * with Hb Change	N Ⱡ	cc with Baseline Hb	N	cc with FU Hb	N
Baseline CRP	91.00 (31.40–152.00)	0.031	59	0.018	84	−0.012	50
FU CRP	2.40 (1.85–6.30)	−0.002	49	NA ‡	50	−0.265	54
Baseline LDH	365.00 (272.00–516.00)	0.082	46	0.012	64	0.089	46
FU LDH	200.00 (170.00–244.75)	0.002	51	NA	51	−0.126	55
Baseline IG	0.03 (0.01–0.08)	0.072	30	0.108	36	0.331	30
FU IG	0.02 (0.01–0.03)	0.075	25	NA	25	−0.006	28
Baseline CPK	102.00 (57.00–298.00)	0.070	34	0.053	52	−0.003	34
FU CPK	72.00 (66.00–100.00)	0.033	31	NA	31	0.084	34
Baseline NLR	5.05 (3.07–7.14)	0.242	46	−0.071	55	0.133	46
FU NLR	1.63 (1.29–2.13)	0.294	36	NA	45	−0.018	49
Baseline D-dimer	0.75 (0.54–2.20)	0.206	55	−0.093	80	−0.030	55

Abbreviations: CRP: C-reactive protein, LDH: Lactate Dehydrogenase, IG: Immature Granulocyte, CPK: creatine phosphokinase, NLR: Neutrophil/Lymphocyte ratio. * Correlation coefficient. Ⱡ number of cases in which correlation analysis between them and Hb change has been performed. ‡ NA: not applicable; our research hypothesis did not consider the relationship between FU inflammatory biomarkers and baseline Hb, since the outcome (Hb concentrations) would precede the exposures in this analysis.

**Table 4 biomedicines-12-01234-t004:** Associations of hemoglobin levels and anemia with fatigue 3–6 months after SARS-CoV-2 infection (average FSS ≥ 4).

	Unadjusted	Adjusted *
	OR ‡ (95% CI)	OR (95% CI)
Baseline Hb levels	0.676 (0.386–1.184)	0.760 (0.430–1.342)
Follow-up Hb levels	0.370 (0.141–0.969)	0.350 (0.120–1.015)
Hb change	0.380 (0.133–1.081)	0.377 (0.130–1.091)
Baseline anemia	1.209 (0.343–4.259)	1.557 (0.420–5.779)
Follow-up anemia	2.143 (0.243–18.919)	2.246 (0.250–20.203)

* Adjusted for age and sex. ‡ Per 1 mmol/L increase in Hb levels.

## Data Availability

The data presented in this study are available upon request to the corresponding author. The data are not publicly available due to agreements made by the consortium, that only allow access by each consortium partner to specific data that answer their pre-specified research questions. A request for access to data by organizations outside of the consortium can be submitted to the P4O2 Data Committee (via p4o2@amsterdamumc.nl) and the research will need to be performed in collaboration with one of the P4O2 consortium partners.

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
