# Peer review of "Hemoglobin and Its Relationship with Fatigue in Long-COVID Patients Three to Six Months after SARS-CoV-2 Infection"

_biomedicines, 2024, doi:10.3390/biomedicines12061234_

Round 1

Reviewer 1 Report

Comments and Suggestions for Authors

Hemoglobin and its relationship with fatigue in Long-COVID 2 patients three to six months after SARS-CoV-2 infection

The authors present an analysis of hemoglobin levels and relationship with fatigue in people suffering with long COVID. The manuscript is interesting and mostly well-written but the statistical analysis is of a poor standard, and appears to include a number of mistakes, or at the very least is unclear.

SPECIFIC COMMENTS

(1) Table 1, line 1, there are 95 participants. 66 have fatigue 3-6 months after infection, 21 have no fatigue. That makes 87, what happened to the other 8?

(2) Table 1, Alfa should be spelled as Alpha

(3) Table 1, according to this table there were N = 47 females (first row), of which 35 were suffering from fatigue (last row). 35 / 47 = 74.5% but the table reports 83.3%, why?

(4) Same point but for males, N = 48, 31 suffered from fatigue, this is 64.6% but the table reports 68.9%, why? I struggled to reconcile all of the % figures in Table 1.

(5) Line 180 reports a p-value of 0.025 for the different proportion of males versus females but does not state the statistical test used (normally chi-square for a difference in proportions) - please state the test.

(6) Same goes for all statistical tests, state the test

(7) Averages are not that helpful in assessing differences between populations, because they do not contain information about variance. Please consider appropriate boxplots. Please add a boxplot in particular for those with fatigue versus those without, for hemoglobin levels. This is your main conclusion "we observed that those with fatigue tended to have lower hemoglobin levels than those without" so warrants a graphical depiction of the data.

(8) Line 227, is causal definitely the right word to use, do the authors have a cause and effect mechanism in mind?

(9) Line 296, I had trouble tracking down the mean hemoglobin value for women (8.84) and men (9.04) reported in Discussion. First, is this statistically significant, what is the p-value, and what test was used? Second, if this is an important number in Discussion, where is it reported in Results? Should the value for women actually be 8.36, as per Table 1?

GENERAL COMMENTS

It is possible that some of the statistical differences are due to missing data, "Hemoglobin levels during the second study visit have not included in this analyses due to the high number of missing data (not being available due to lost to follow up as well as not performing clinic visit at the late stage of inclusion)". This sentence is confusingly written. Table 1 seems to suggest data were missing for 37 cases.

I strongly suggest the paper is rewritten and participants with incomplete data are excluded altogether from the study and the statistics can then be calculated consistently on that basis. Including participants in some calculations and not others introduces bias and prevents comparisons of trends.

Alternatively, if each calculation is done with a different population, such that % values change from row to row, the data on the total N for each calculation must be given explicitly, or the reader cannot reproduce your calculations. This is problematic for obvious reasons, an inability to calculate the same % values or p-values undermines the credibility of your manuscript.

Comments on the Quality of English Language

The English is mostly fine with minor issues that can be picked up in the proof edit.

Author Response

We thank the reviewer for the useful comments and suggestions. We have adjusted the manuscript accordingly and responses to each comment are provided in the file submitted below.

Reviewer 1:

We thank the reviewer for the useful comments and suggestions. We have adjusted the manuscript accordingly and responses to each comment are provided as below (in red).

Comments and Suggestions for Authors

Hemoglobin and its relationship with fatigue in Long-COVID 2 patients three to six months after SARS-CoV-2 infection

The authors present an analysis of hemoglobin levels and relationship with fatigue in people suffering with long COVID. The manuscript is interesting and mostly well-written but the statistical analysis is of a poor standard, and appears to include a number of mistakes, or at the very least is unclear.

SPECIFIC COMMENTS

(1) Table 1, line 1, there are 95 participants. 66 have fatigue 3-6 months after infection, 21 have no fatigue. That makes 87, what happened to the other 8? --> For these 8 cases, fatigue data was missing, Table one has been updated to better clarify this

(2) Table 1, Alfa should be spelled as Alpha --> the spelling is now corrected in the manuscript

(3) Table 1, according to this table there were N = 47 females (first row), of which 35 were suffering from fatigue (last row). 35 / 47 = 74.5% but the table reports 83.3%, why? --> The original percentage calculation excluded the 5 missing female and 3 missing male participants. This has now been updated to include those with missing fatigue information.

(4) Same point but for males, N = 48, 31 suffered from fatigue, this is 64.6% but the table reports 68.9%, why? I struggled to reconcile all of the % figures in Table 1. --> The original percentage calculation excluded the 5 missing female and 3 missing male participants. This has now been updated to include those with missing fatigue information.

(5) Line 180 reports a p-value of 0.025 for the different proportion of males versus females but does not state the statistical test used (normally chi-square for a difference in proportions) - please state the test. --> We used Chi-square test for comparing fatigue between men and women and it has been now mentioned in the manuscript too.

(6) Same goes for all statistical tests, state the test --> All performed statistical tests have now been mentioned.

(7) Averages are not that helpful in assessing differences between populations, because they do not contain information about variance. Please consider appropriate boxplots. Please add a boxplot in particular for those with fatigue versus those without, for hemoglobin levels. This is your main conclusion "we observed that those with fatigue tended to have lower hemoglobin levels than those without" so warrants a graphical depiction of the data. --> A figure containing Boxplots has been added to the manuscript (after table 2)

(8) Line 227, is causal definitely the right word to use, do the authors have a cause and effect mechanism in mind?  --> When generating our DAG, we considered a potentially causal relationship between Hb and fatigue. However, we are unable to analytically prove such a relationship in the current paper. We have therefore revised the use of the term in the manuscript.

(9) Line 296, I had trouble tracking down the mean hemoglobin value for women (8.84) and men (9.04) reported in Discussion. First, is this statistically significant, what is the p-value, and what test was used? Second, if this is an important number in Discussion, where is it reported in Results? Should the value for women actually be 8.36, as per Table 1? --> Unfortunately, the value reported in the discussion was a typo and 8.36 is the correct amount for mean Hb level in women during the long COVID phase. This has now been corrected in the discussion. This has also been added to the results section as the following: “Hemoglobin values were significantly different (via Independent Samples Test) between men and women both during the acute phase (8.84 vs 8.35 mmol/L respectively p-value=0.023) and at follow-up (9.04 vs 8.36 mmol/L p-value=0.003,) (Table 2).”

GENERAL COMMENTS

It is possible that some of the statistical differences are due to missing data, "Hemoglobin levels during the second study visit have not included in this analyses due to the high number of missing data (not being available due to lost to follow up as well as not performing clinic visit at the late stage of inclusion)". This sentence is confusingly written. Table 1 seems to suggest data were missing for 37 cases.-->The “second” visit referred to is in reference to the 12-18 month follow-up visit, which was not included in the current study. To resolve any confusion, we have changed the sentence in the manuscript as follows: “The second study visit was conducted 12-18 months after infection, which included fairly high number of lost to FU among whom only 11 provided blood samples for hemoglobin”. 

I strongly suggest the paper is rewritten and participants with incomplete data are excluded altogether from the study and the statistics can then be calculated consistently on that basis. Including participants in some calculations and not others introduces bias and prevents comparisons of trends.

Alternatively, if each calculation is done with a different population, such that % values change from row to row, the data on the total N for each calculation must be given explicitly, or the reader cannot reproduce your calculations. This is problematic for obvious reasons, an inability to calculate the same % values or p-values undermines the credibility of your manuscript. --> Following internal discussion, we have decided to include all cases that have at least one Hb datapoint (first study visit/baseline) available in order to study as much data as possible. However, in order to make the results clearer, we have now included counts of all missing datapoints in their relevant tables and tests.

Comments on the Quality of English Language

The English is mostly fine with minor issues that can be picked up in the proof edit.

Submission Date

11 February 2024

Date of this review

22 Feb 2024 13:52:42

Reviewer 2 Report

Comments and Suggestions for Authors

The introduction presents a comprehensive overview of the long-term consequences of SARS-CoV-2 infections, particularly focusing on the potential association between decreased hemoglobin levels and Long-COVID symptoms.

·       The statement "Three years since the World Health Organization (WHO) officially announced the COVID-19 pandemic" is inaccurate. As of now, the COVID-19 pandemic has not been ongoing for three years; the first case was reported in late 2019.

·       The introduction lacks a clear transition between the general introduction of Long-COVID and the specific focus on decreased hemoglobin levels. A more seamless transition would enhance the overall flow.

·       The introduction introduces the term "Long-COVID" without providing a clear definition, which might be confusing for readers unfamiliar with the term. A brief definition or explanation would be helpful

·       The proposed mechanism of hemoglobin conversion into methemoglobin (MetHb) and carboxyhemoglobin (COHb) during the acute phase of SARS-CoV-2 infection needs further scientific support and clarification. The pathways described are oversimplified, and the connection to hemolytic anemia is not well-established.

·       The introduction references studies (8, 9) without providing the specific details or context of these studies. Including more information about these studies and their findings would enhance the credibility of the information presented.

·       The mention of decreased hemoglobin levels and its association with fatigue is repeated in several instances. Redundancy should be minimized, and key points should be reinforced without excessive repetition.

·       The introduction lacks a clear and concise thesis statement outlining the main purpose and objective of the study. Clearly stating the research question or hypothesis would provide a better roadmap for readers.

·       The tone is generally formal, but some sentences are complex and may benefit from simplification for better clarity. Additionally, ensuring precise language use will enhance the overall quality of the introduction.

Methods:

The method section provides a detailed overview of the study design, participant selection, ethical considerations, variables, and statistical analysis.

·       The criteria for participant inclusion and exclusion lack clarity. For instance, the rationale behind selecting participants aged 40-65 years and the specific criteria for excluding participants during the second study visit need further elaboration.

·       The exclusion of hemoglobin levels during the second study visit due to a high number of missing data raises concerns about potential bias. Providing more information on the reasons for missing data and considering methods to address this issue (e.g., imputation techniques) would enhance the robustness of the analysis.

·       The definition of anemia is provided, but the reasoning behind choosing the specific hemoglobin levels (less than 7.5 mmol/L in women and less than 8.5 mmol/L in men) is not adequately explained. Providing a clear justification for these cutoffs is essential for the validity of the results.

·       The classification of hemoglobin changes into increased, decreased, or unchanged is mentioned, but the criteria for determining the clinical significance of these changes (i.e., 10% or more) lack a strong scientific basis. The reference to obstetrics and gynecology standards does not seem directly relevant to COVID-19 and Long-COVID patients, and a more context-specific justification is needed.

·       While the FSS is mentioned, the rationale for choosing a cutoff of four for categorizing patients into those with or without fatigue is not explained. Providing a clear justification for this cutoff would strengthen the validity of the fatigue classification.

·       The selection of inflammatory biomarkers is justified, but there is no discussion about the specific relevance of each biomarker to fatigue or hemoglobin disruption. Adding a brief explanation of the rationale behind choosing these biomarkers would enhance the scientific rigor of the study.

·       The statistical analysis is well-detailed, but there is no mention of addressing multiple comparisons, which is important to control the overall Type I error rate. A brief comment on how multiple comparisons were handled would be beneficial.

·       The sensitivity analysis based on renal function is a valuable addition. However, the choice of a GFR greater than 60 mL/min/1.73m2 as a cutoff point lacks sufficient justification. Providing a rationale for this specific cutoff is necessary.

·       The version of SPSS used is mentioned, but details about specific procedures, tests, and adjustments made within the software are not provided. Additionally, the choice of a significance level (p-value ≤ 0.05) is standard but merits a brief acknowledgment.

Results:

·       The reported p-values for differences in fatigue between males and females (p-value = 0.025) and BMI differences (p-value = 0.002) are mentioned but not provided in Table 1. Transparency in presenting statistical values is crucial for the readers to assess the significance of these differences.

·       The reporting of hemoglobin levels lacks consistency. For instance, the mean hemoglobin concentration is mentioned as 8.6 mmol/L for the entire study population, but later in the text, it is stated as 8.61 mmol/L during the acute phase. Such discrepancies need clarification for accurate interpretation.

·       The reported prevalence of anemia during the acute phase (22.6%) and Long-COVID phase (16.4%) is not consistently supported by the provided data. The breakdown of cases (e.g., seven continuing anemia, three new-onset) lacks clarity and completeness.

·       The comparison of hemoglobin levels between men and women and the subsequent analysis of changes in hemoglobin do not reach statistical significance (p-value=0.222). The text implies a trend toward significance, and this should be accurately communicated to avoid misinterpretation.

·       The lack of strong correlations between hemoglobin levels and inflammatory biomarkers is reported, but the specific biomarkers and their relevance to the study are not discussed. Providing a brief rationale for the chosen biomarkers would enhance the scientific context.

·       The text mentions adjustments made for age and sex in the logistic regression model due to the small sample size but does not provide details about the nature of these adjustments. Clearly stating how age and sex were accounted for in the model is essential for transparency.

·       The reported associations between hemoglobin levels and fatigue lack clarity in interpretation. For example, the adjusted ORs for hemoglobin levels 3-6 months after infection are given as 0.35 and 0.38, but the interpretation of these values is not provided. Clarifying the direction and strength of these associations is crucial for understanding the impact of hemoglobin on fatigue.

·       The sensitivity analysis excluding cases with impaired renal function is mentioned, but the choice of a GFR cutoff (60 mL/min/1.73m2) lacks sufficient justification. Providing a rationale for this specific cutoff would strengthen the validity of the sensitivity analysis.

Discussion and conclusion:

The discussion provided discusses the role of hemoglobin levels during and after the acute phase of SARS-CoV-2 infection in relation to inflammatory biomarkers and fatigue in Long-COVID patients.

·       The discussion mentions that 16.41% of participants were anemic 3-6 months after SARS-CoV-2 infection, but it doesn't define the criteria used for diagnosing anemia. Different criteria might lead to variations in prevalence rates.

·       The study should clearly state the hemoglobin thresholds used to define anemia and discuss how these thresholds were determined.

·       The discussion suggests a possible role of anemia, either pre-existing or caused by acute infection, in contributing to Long-COVID fatigue. However, the study design is observational, and causation cannot be firmly established based on correlations alone.

·       It's essential to emphasize the observational nature of the study and the need for further research to establish causation.

·       The study acknowledges limitations in statistical power, particularly in regression analysis, due to a limited sample size. This raises concerns about the reliability and generalizability of the findings.

·       The missing values for baseline variables, especially inflammatory biomarkers, further compromise the study's ability to draw robust conclusions.

·       The study fails to find correlations between hemoglobin and certain inflammatory biomarkers, but the discussion doesn't thoroughly explore potential reasons for this lack of correlation.

·       The study should provide a more detailed discussion on the selection of inflammatory biomarkers and their relevance to the research question.

·       The study's age range of 40 to 65 years limits the generalizability of findings to other age groups. It would be helpful to discuss how the chosen age range was determined and its implications for broader applicability.

·       The discussion introduces the association between hemoglobin and oxidative stress without delving into the specific mechanisms or providing a comprehensive literature review on this topic.

·       The study should elaborate on the relationship between hemoglobin and oxidative stress, providing a more thorough background.

·       The study concludes with potential clinical implications without providing concrete recommendations for monitoring or treating hemoglobin levels in Long-COVID patients.

·       The discussion should include more specific recommendations for clinicians based on the study's findings and limitations.

Comments on the Quality of English Language

 Extensive editing of English language required

Author Response

We thank the reviewer for the useful comments and suggestions. We have adjusted the manuscript accordingly and responses to each comment are provided in the file submitted below.

Reviewer 3 Report

Comments and Suggestions for Authors

The article entitled as "Hemoglobin and its relationship with fatigue in Long-COVID 2 patients three to six months after SARS-CoV-2 infection" is an interesting piece of work. 

The researchers observed that patients with fatigue tended to have lower hemoglobin levels than those without. The present findings suggest the important role of hemoglobin in Long-COVID fatigue.

The findings are interesting and well presented. 

I have some minor suggestions: 

If possible, provide the more relevance of this present study while the emergence of plethora research studies on Long COVID.

Further, improving the discussion with more concrete evidence/recent references. 

Rewrite the conclusion section as a separate section. 

Best Wishes

Comments on the Quality of English Language

The language is fine, final checks are required to avoid any grammatical errors. 

Author Response

We thank the reviewer for the useful comments and suggestions. We have adjusted the manuscript accordingly and responses to each comment are provided in the file submitted below.

Reviewer 2:

We thank the reviewer for their useful comments and suggestion; we have adjusted the manuscript accordingly and response to each comment is provided as below (in red).

Comments and Suggestions for Authors

The introduction presents a comprehensive overview of the long-term consequences of SARS-CoV-2 infections, particularly focusing on the potential association between decreased hemoglobin levels and Long-COVID symptoms.

  • The statement "Three years since the World Health Organization (WHO) officially announced the COVID-19 pandemic" is inaccurate. As of now, the COVID-19 pandemic has not been ongoing for three years; the first case was reported in late 2019. --> this has been now changed in the manuscript as follows; “Over four years ago, the World Health Organization (WHO) declared COVID-19 a global pandemic.”
  • The introduction lacks a clear transition between the general introduction of Long-COVID and the specific focus on decreased hemoglobin levels. A more seamless transition would enhance the overall flow. --> an additional paragraph has been added between to better transition between these two subjects, providing more details on Long-COVID, as well as focusing on (less considered) non-respiratory consequences: “Long-COVID is causing widespread disability worldwide, with many potentially developing lifelong disabilities. Nevertheless, current diagnostics and treatments are insufficient, necessitating urgent clinical investigations to address hypothesized underlying biological mechanisms (9). Whilst the mechanism of long-COVID continues to be unclear due to the wide clinical spectrum and organ involvement, only a thorough understanding of its pathophysiology will allow us to assess and treat the disease's side effects (10). While persistence of respiratory symptoms, particularly dyspnea and cough, appear to be common in Long-COVID, some of the other persistent symptoms of Long-COVID may be part of a multisystem disease, and interdisciplinary treatment of these chronic sequelae is necessary (11).”
  • The introduction introduces the term "Long-COVID" without providing a clear definition, which might be confusing for readers unfamiliar with the term. A brief definition or explanation would be helpful --> now this comment is already addressed as mentioned above, also, further explanations regarding the typical symptoms has been now added to the manuscript as following: “The term “Long-COVID” refers to the condition in which patients experience a variety of persisting symptoms long after the acute phase of the infection has resolved but often typified by fatigue, dyspnea, brain fog, sleep problems, muscle stiffness, and mental problems”
  • The proposed mechanism of hemoglobin conversion into methemoglobin (MetHb) and carboxyhemoglobin (COHb) during the acute phase of SARS-CoV-2 infection needs further scientific support and clarification. The pathways described are oversimplified, and the connection to hemolytic anemia is not well-established. --> Additional explanations were added to the manuscript as follows: “). While oxidative stress, which is linked to infectious diseases, may contribute (15), medications given during treatment may also be relevant, given that certain medicines such as hydroxychloroquine used to treat COVID-19 have a high potential for oxidation, which can lead to MetHb and COHb (16).”
  • The introduction references studies (8, 9) without providing the specific details or context of these studies. Including more information about these studies and their findings would enhance the credibility of the information presented. --> Additional explanations regarding these studies details were added as following (to avoid too much repetition of reference number 9, a new reference has been added): “Based on the results of multivariate regression of a study comprising 1147 COVID-19 patients across five tertiary pediatric hospitals in Asia, higher levels of hemoglobin and platelets were protective against severe/critical COVID-19 infection (8). A retrospective investigation of 23 COVID-19 hospitalized patients with pneumonia also revealed a significant association between higher levels of hemoglobin and a reduced need for mechanical ventilation administration, with an odds ratio of 0.313 (13).”
  • The mention of decreased hemoglobin levels and its association with fatigue is repeated in several instances. Redundancy should be minimized, and key points should be reinforced without excessive repetition. --> this has now been considered while doing the English revision
  • The introduction lacks a clear and concise thesis statement outlining the main purpose and objective of the study. Clearly stating the research question or hypothesis would provide a better roadmap for readers. --> The study objective as stated in our original submission was as follows: “The purpose of the current study is therefore to examine the correlations between hemoglobin levels and inflammatory biomarkers, and to study the association between hemoglobin and fatigue in a cohort of Long-COVID patients.” We have added further elaboration as follows; “The purpose of the current study is therefore to examine the correlations between hemoglobin levels and inflammatory biomarkers and to evaluate the potential role of inflammatory process in decreasing hemoglobin levels in Long-COVID patients. Furthermore, this study investigates the association between hemoglobin and fatigue in a cohort of Long-COVID patients to evaluate whether hemoglobin levels contribute to developing fatigue in these patients.”
  • The tone is generally formal, but some sentences are complex and may benefit from simplification for better clarity. Additionally, ensuring precise language use will enhance the overall quality of the introduction. --> the text was reviewed by a native English author prior to resubmission.

Methods:

The method section provides a detailed overview of the study design, participant selection, ethical considerations, variables, and statistical analysis.

  • The criteria for participant inclusion and exclusion lack clarity. For instance, the rationale behind selecting participants aged 40-65 years and the specific criteria for excluding participants during the second study visit need further elaboration. --> The cohort was started at a point in time when not much was known about Long-COVID and therefore no specific age group was known to be important at that time Because older patients (above 65) are more likely to have more comorbidities that may interfere with the analysis we decided to limit our population to below that age. The previous sentence has been now added to the discussion section (further explanation regarding the limitations and age range). Also, the following part has been added to the method section regarding the exclusion criteria; “There was no specific exclusion criteria for the FU study visit of this cohort, and nine months following the initial study visit, all participants received invitations for a follow-up appointment, but a portion was lost to follow-up.”
  • The exclusion of hemoglobin levels during the second study visit due to a high number of missing data raises concerns about potential bias. Providing more information on the reasons for missing data and considering methods to address this issue (e.g., imputation techniques) would enhance the robustness of the analysis. --> the second study visit was conducted 12-18 months after infection, which included fairly high number of lost to FU among whom only 11 provided blood samples for hemoglobin. Imputation was not considered, as Hb was the primary variable of interest in this study.
  • The definition of anemia is provided, but the reasoning behind choosing the specific hemoglobin levels (less than 7.5 mmol/L in women and less than 8.5 mmol/L in men) is not adequately explained. Providing a clear justification for these cutoffs is essential for the validity of the results. à these cut off points were used as these were the reference range of the measuring laboratory
  • The classification of hemoglobin changes into increased, decreased, or unchanged is mentioned, but the criteria for determining the clinical significance of these changes (i.e., 10% or more) lack a strong scientific basis. The reference to obstetrics and gynecology standards does not seem directly relevant to COVID-19 and Long-COVID patients, and a more context-specific justification is needed. --> As relatively little was known regarding associations between hemoglobin and fatigue in the Long-Covid setting, we sought to identify a change in Hb which might be clinically relevant, even if not resulting in anemia. Therefore, based on routine clinical practice, a more than 10% decline in Hb (approximately 1 mmol/L), would be considered a significant Hb drop, warranting further analysis. However, as we mention in the article, there is not a universal consensus surrounding this among internal medicine papers. Instead, we refer to the GYN field, where a stronger consensus is presented.
  • While the FSS is mentioned, the rationale for choosing a cutoff of four for categorizing patients into those with or without fatigue is not explained. Providing a clear justification for this cutoff would strengthen the validity of the fatigue classification. --> the FSS cut off point is different for various populations. In our P4O2 Long-COVID cohort we have consistently chosen to use 4 as a cutoff point. We have also added the following explanations to the manuscript :” The fatigue severity scale originally utilized a cut-off of 4 or more to indicate severe fatigue. We acknowledge that different cut-offs have been developed, including low (<4), medium/borderline (4 - 5), and high/severe (>5) (32). However, for simplicity, and consistency with our other work in this population (27) we have chosen to consistently use 4 as a cut-off point”
  • The selection of inflammatory biomarkers is justified, but there is no discussion about the specific relevance of each biomarker to fatigue or hemoglobin disruption. Adding a brief explanation of the rationale behind choosing these biomarkers would enhance the scientific rigor of the study. --> In our analysis, we sought to evaluate whether any inflammatory process would be associated with hemoglobin. The biomarkers present in the current study reflect the total available inflammatory indices obtained from their records in the clinical care. The following sentence, as well as references indicating these factors are inflammatory biomarkers have been added to justify this: “Blood samples were also examined for a variety of biomarkers, and markers which indicated the presence of an inflammatory process were included as inflammatory indices in the current study. Identified markers were: creatinine, D-Dimer, Neutrophil/Lymphocyte ratio (NLR), Immature Granulocyte (IG), C-reactive protein (CRP), Creatine phosphokinase (CPK), and Lactate Dehydrogenase (LDH) (33 - 35). Normal ranges and analytical approaches were as per the standard operating procedures at participating hospitals.”
  • The statistical analysis is well-detailed, but there is no mention of addressing multiple comparisons, which is important to control the overall Type I error rate. A brief comment on how multiple comparisons were handled would be beneficial. --> Bonferroni correction was considered for reducing Type 1 error in our multiple comparisons – especially with regard to inflammatory markers. However, no statistically significant result was found, even without correction, for the inflammatory markers. Thus, no additional correction was performed. This sentence has been also added to the manuscript as following: “Bonferroni correction was considered given the multiple correlation testing, however as no statistically significant result was found no further correction was undertaken”

       The sensitivity analysis based on renal function is a valuable addition. However, the choice of a GFR greater than 60 mL/min/1.73m2 as a cutoff point lacks sufficient justification. Providing a rationale for this specific cutoff is necessary. --> we selected a GFR of 60 as this represents the cut off point for mild kidney injury, and therefore less likely for individuals to have renally induced anemia. We have added the following explanatory sentence: ” because this is the cut-off point that differentiates cases of mild chronic kidney disease from more severe cases where the prevalence of anemia is higher (37-39).”

  • The version of SPSS used is mentioned, but details about specific procedures, tests, and adjustments made within the software are not provided. Additionally, the choice of a significance level (p-value ≤ 0.05) is standard but merits a brief acknowledgment. --> additional explanations have been now added to the related text: “The Paired-Samples T test was utilized to compare parametric variables between two time points (baseline and first FU visit) and the Independent Samples Test was used to compare parametric variables between two different groups (fatigue patients vs non-fatigue patients, male vs female). Chi-Square testing was employed while com-paring categorical variables between these groups, and for comparing non-parametric variables between them Mann-Whitney U test was used.” and “All statistical analyses were performed by SPSS version 28 and a p-value ≤0.05 was considered statistically significant for all statistical tests.”

Results:

  • The reported p-values for differences in fatigue between males and females (p-value = 0.025) and BMI differences (p-value = 0.002) are mentioned but not provided in Table 1. Transparency in presenting statistical values is crucial for the readers to assess the significance of these differences. --> We have now added p-values to Table 1 as requested. adding a p value column to table 1 was not considered, as this was actually a descriptive table and just for only a few variables p-values were calculated. However, because of transparency we have now added all p-values in Table 1.
  • The reporting of hemoglobin levels lacks consistency. For instance, the mean hemoglobin concentration is mentioned as 8.6 mmol/L for the entire study population, but later in the text, it is stated as 8.61 mmol/L during the acute phase. Such discrepancies need clarification for accurate interpretation. --> This inconsistency has been corrected.
  • The reported prevalence of anemia during the acute phase (22.6%) and Long-COVID phase (16.4%) is not consistently supported by the provided data. The breakdown of cases (e.g., seven continuing anemia, three new-onset) lacks clarity and completeness. --> Owing to incomplete Hb data for either baseline or FU visit we were unable to calculate the new-onset FU anemia/ persistent anemia/cured anemia for all cases. We have added a table (table S1) to the supplement to better illustrate the distribution of anemia at baseline and follow-up, including those with incomplete data.

Table S1. Frequency and persistence of anemia in the study population

Baseline anemia

19

Cured in FU (baseline anemia but no FU anemia)

8/19

Persistent in FU ‡

7/19

Unknown (follow-up information missing)Ⱡ

4/19

FU anemia

11

New onset anemia (no anemia at baseline)

3/11

Persistent in FU  ‡

7/11

Unknown (baseline information missing)Ⱡ

1/11

‡ a computed variable; baseline anemia =1 (Yes), if FU anemia = 1 (Yes)

Ⱡ either baseline anemia data or FU anemia data were missing

  • The comparison of hemoglobin levels between men and women and the subsequent analysis of changes in hemoglobin do not reach statistical significance (p-value=0.222). The text implies a trend toward significance, and this should be accurately communicated to avoid misinterpretation. --> It has been now changed in the manuscript as follows: “However, men and women's differences in Hemoglobin change did not reach statistical significance (p-value=0.222 according to Independent Samples Test). ”
  • The lack of strong correlations between hemoglobin levels and inflammatory biomarkers is reported, but the specific biomarkers and their relevance to the study are not discussed. Providing a brief rationale for the chosen biomarkers would enhance the scientific context. --> the purpose of biomarker selection was to examine inflammatory processes and changes in hemoglobin. This has now been added to the methods section as below: “Blood samples were also examined for a variety of biomarkers, and markers which indicated the presence of an inflammatory process were included as inflammatory in-dices in the current study. Identified markers were: creatinine, D-Dimer, Neutrophil/Lymphocyte ratio (NLR), Immature Granulocyte (IG), C-reactive protein (CRP), Creatine phosphokinase (CPK), and Lactate Dehydrogenase (LDH). Normal ranges and analytical approaches were as per the standard operating procedures at participating hospitals.”
  • The text mentions adjustments made for age and sex in the logistic regression model due to the small sample size but does not provide details about the nature of these adjustments. Clearly stating how age and sex were accounted for in the model is essential for transparency. --> after making a DAG to identify potential confounders, adjustment was performed by running a model where age and sex were added as covariates. As a rule of thumb, we would ideally desire considering 20 events per covariate when generating our models. Given the relatively small sample size here, we opted to include age and sex, as the most important covariates, in our model. Additional explanations have also been added to the manuscript as follows: “According to this DAG, age, sex, ethnicity, BMI, underlying diseases, hospital admission, ICU admission, inflammation, and lung function were all identified confounders for the potential pathway of developing fatigue due to hemoglobin alteration. However, considering the sample size and a target of 20 events per covariate, only age and sex, identified as the most important potential confounders, were included in the regression analysis.”
  • The reported associations between hemoglobin levels and fatigue lack clarity in interpretation. For example, the adjusted ORs for hemoglobin levels 3-6 months after infection are given as 0.35 and 0.38, but the interpretation of these values is not provided. Clarifying the direction and strength of these associations is crucial for understanding the impact of hemoglobin on fatigue. --> further elaborations have now been added to the related text to make clarifications: “ 4). While findings were non-significant, an association between increased Hb, and reduced fatigue was consistently observed. The strongest associations were observed with hemoglobin levels 3-6 months after infection where increased hemoglobin was associated with a reduced likelihood of fatigue. A 1 mmol/L increase hemoglobin was associated with an adjusted odds ratio of 0.35 for fatigue (95%CI 0.12 - 1.02). Another strong relationship between fatigue and hemoglobin indices was observed for the change in hemoglobin levels, where a 1mmol/L positive increase in Hb (i.e. greater in visit 2 than visit 1) resulted in an adjusted of OR 0.38 (95% CI 0.13 - 1.09). There was also a positive association between fatigue and anemia at both baseline and at follow-up (adjusted OR 1.56 [95% CI 0.42 – 5.78] and OR 2.25 [95% CI 0.25 - 20.20], respectively).”
  • The sensitivity analysis excluding cases with impaired renal function is mentioned, but the choice of a GFR cutoff (60 mL/min/1.73m2) lacks sufficient justification. Providing a rationale for this specific cutoff would strengthen the validity of the sensitivity analysis. --> we selected a GFR of 60 as this represents the cut off point for mild kidney injury, and therefore less likely for individuals to have renally induced anemia. We have added the following explanatory sentence: ” because this is the cut-off point that differentiates cases of mild chronic kidney disease from more severe cases where the prevalence of anemia is higher (37 - 39).”

Discussion and conclusion:

The discussion provided discusses the role of hemoglobin levels during and after the acute phase of SARS-CoV-2 infection in relation to inflammatory biomarkers and fatigue in Long-COVID patients.

  • The discussion mentions that 16.41% of participants were anemic 3-6 months after SARS-CoV-2 infection, but it doesn't define the criteria used for diagnosing anemia. Different criteria might lead to variations in prevalence rates.--> Anemia was defined as an Hb level less than 7.5 mmol/dl in women and less than 8.5mmol/dl in men. This definition was based upon the testing laboratory’s reference range, and an explanation has already added to the methods section (see related comment on the method section)
  • The study should clearly state the hemoglobin thresholds used to define anemia and discuss how these thresholds were determined. à this comment is already addressed above
  • The discussion suggests a possible role of anemia, either pre-existing or caused by acute infection, in contributing to Long-COVID fatigue. However, the study design is observational, and causation cannot be firmly established based on correlations alone. --> we completely agree with this comment and have added a sentence to make this even more clear: “Nevertheless, since the current study is observational in nature, further investigation is necessary to establish causation.”.
  • It's essential to emphasize the observational nature of the study and the need for further research to establish causation. --> as mentioned above, this has been now added to the manuscript
  • The study acknowledges limitations in statistical power, particularly in regression analysis, due to a limited sample size. This raises concerns about the reliability and generalizability of the findings. --> this comment has been now incorporated in strength/limitation section (issues about generalizability has been already addressed in the last part after mentioning the age rang): and limits the external generalizability of our findings”
  • The missing values for baseline variables, especially inflammatory biomarkers, further compromise the study's ability to draw robust conclusions. --> this point has been now highlighted in conclusion and following sentence has been now added there: “Particularly as it is more challenging for the current study to draw definitive conclusions where some variables, most notably inflammatory biomarkers, are missing. ”
  • The study fails to find correlations between hemoglobin and certain inflammatory biomarkers, but the discussion doesn't thoroughly explore potential reasons for this lack of correlation. --> additional explanation has been now added as potential reasons for not observing correlations between Hb and our inflammatory biomarkers.: “This difference may illustrate that only extremely severe inflammation can affect hemoglobin values, that our sample size was insufficient to identify an effect, or that the inflammatory biomarkers available for use in the current study were not optimal for the evaluation of inflammatory driven Hb disruption. For example, the erythropoiesis-suppressing effects of specific inflammatory cytokines, such as TNFα, IL-1, and interferon-γ, have been reported in various investigations. Therefore, a broader assessment of inflammatory markers alongside other diagnostic tests for anemia related to inflammation (serum iron, ferritin, and hepcidin), may be a more accurate way to assess the correlation between anemia and inflammation (49). For example, in an analysis of 214 ex COVID-19 patients, where all aforementioned variables were included, patients who reported long-COVID Syndrome months after the acute phase were distinguished by inflammation and abnormal iron homeostasis that lasted longer than two weeks after the onset of symptoms. This led the researchers to conclude that the effects of inflammatory iron dysregulation on erythropoiesis and blood oxygen transportation may be partially responsible for several of the characteristics of Long-COVID (50). Alternately this might suggest that factors other than inflammation have a more significant role. For example, there is evidence that SARS-CoV-2 infection can produce hemolytic anemia through either generating structural changes in host cells (RBCs) via viral invasion or activating auto antibodies (51).”
  • The study should provide a more detailed discussion on the selection of inflammatory biomarkers and their relevance to the research question. --> These biomarkers were actually the available inflammatory factors in our dataset for now (and for baseline data, regarding the retrospective method of extraction, these were the only available ones) and we included them in the analysis as they can indicate presence of inflammation in the patients’ body to answer our research question regarding the potential role of inflammation in developing anemia. Further elaboration as mentioned in the comment above has been added to the discussion to further elaborate on this.
  • The study's age range of 40 to 65 years limits the generalizability of findings to other age groups. It would be helpful to discuss how the chosen age range was determined and its implications for broader applicability. --> further elaboration regarding the age range has been now mentioned in the methodology section comments and now has been also added to the discussion as following:” Finally, this cohort was started at a point in time when not much was known about Long-COVID and therefore no specific age group was known to be important at that time, and it seemed that it was mainly affecting individuals from 40 years and older. In addition, because older patients (above 65) are more likely to have more comorbidities that may interfere with the analysis we decided to limit our population to below that age. Thus, the age range of our study population (40 to 65 years) makes us more cautious in generalizing our findings to other age ranges.”
  • The discussion introduces the association between hemoglobin and oxidative stress without delving into the specific mechanisms or providing a comprehensive literature review on this topic. ·       The study should elaborate on the relationship between hemoglobin and oxidative stress, providing a more thorough background. --> further elaboration regarding the mechanism through which oxidative stress affects Hb function has been now explained in the manuscript and following sentences has been added to the text: “Experiments have shown that oxidative stress significantly impacts the structure of hemoporphyrin, which in turn affects hemoglobin function and reduces the efficiency with which hemoglobin binds to oxygen. This process is caused by lipid peroxidation (LPO), which is followed by a  formation of LPO's primary and end products.”
  • The study concludes with potential clinical implications without providing concrete recommendations for monitoring or treating hemoglobin levels in Long-COVID patients. ·       The discussion should include more specific recommendations for clinicians based on the study's findings and limitations --> due to the observational nature of the current study, no strong clinical recommendation could be driven from our findings, however a potential clinical application is that physicians’ continue to monitor Hb in their Long-COVID patients. To epmhasise this, the following has been added to the manuscript: “Our study findings have potential clinical implications, providing evidence for physicians to be mindful about monitoring the hemoglobin level in their Long-COVID patients, particularly in those suffering from fatigue.”

Comments on the Quality of English Language

 Extensive editing of English language required

Submission Date

11 February 2024

Date of this review

20 Feb 2024 20:36:17

Bottom of Form

© 1996-2024 MDPI (Basel, Switzerland) unless otherwise stated

Round 2

Reviewer 1 Report

Comments and Suggestions for Authors

Hemoglobin and its relationship with fatigue in Long-COVID 2 patients three to six months after SARS-CoV-2 infection

The authors have responded to the comments on statistical inconsistencies, and a number of other issues; I thank the authors for their substantial efforts. With the manuscript on a statistically more sound footing, I believe it is close to being suitable for publication. There are, however, a number of other issues which could helpfully be improved, mainly related to Discussion.

(1) References 1 and 2 may not be the most up-to-date citations - the authors state that four years after the WHO declared the pandemic "infections from SARS-CoV-2 and their long-term consequences remain a global challenge (1, 2)" but the cited references are from 2020 and 2021. It might be better to use current literature references. I suggest PMID: 36639608 and PMID: 36620963 as more recent examples, that are more focused on the challenges posed by your specific research.

(2) Line 117, "Finding biomarkers that can assist these patients receive optimized care is crucial, and since hemoglobin can be easily to assess, is it a feature that correcting that could be a possible treatable trait." would be better written as "Finding biomarkers that can help to direct optimized care to patients most at risk is crucial, and since hemoglobin is easily assessed by assay and has been associated with long-COVID, we believe that it is a biomarker worthy of further investigation"

(3) Line 162, missing full stop

(4) Line 205, I'm not sure I follow the logic. A better reason not to use Bonferroni would be that false-discovery correction when testing small numbers of variables which are themselves correlated can be too conservative, as the effective number of tests undertaken (hemoglobin versus inflammation) is not the same as the actual number of tests undertaken (hemoglobin versus x different measures of inflammation, which are clearly not x independent tests). For your purposes, it is probably sufficient to state that FDR was not used as it can be overly conservative especially when validating a smaller number of previously described features or in cases where the effective number of tests is smaller (due to redundancy in similar features measuring the same latent variables) and referencing PMID: 16077740 and PMID: 21451529.

(5) Figure labelling, both figures are currently captioned as fig.2

(6) The boxplot figure would be better presented as two boxplots next to each other (with larger text to keep them readable), the current boxplots take up a lot of page space without conveying any extra information than if they were next to each other 

(7) Lines 431 onwards referencing Hanson et al from 2024 (PMID: 38429458) presents a very interesting point, which I think the authors should expand upon, whereby Hanson et al found a link between inflammation and Hb but your analysis did not validate this link. COVID-19 is an illness which in its short life has already seen many variants and changes in presentation, which in turn increases the risk that correlations or biomarkers seen in one dataset may not be reproducible in other cohorts, as seen in PMID: 37762673. This emphasises the need for validation work to test the robustness of biomarker and pathway analyses, which you have done here.

(8) There may also be some value in referencing post-influenza conditions, and that this is a limitation of the study (are these findings specific to COVID, or indicative of post-viral syndromes more broadly). PMID: 38104583 may be helpful in this context.

(9) In my view, the Conclusions section could be slightly expanded to mention the importance of validation studies to test whether initial findings are reproducible and robust across different cohorts.

Comments on the Quality of English Language

No major comments.

Author Response

Dear Reviewer,

We thank the reviewer for their interesting suggestions and valuable comments. We have adjusted the manuscript accordingly and responses to all comments have been provided in the attached fin (in red font).

Kind regards,

Somayeh Bazdar 

Dear Reviewer,

We thank the reviewer for their interesting suggestions and valuable comments. We have adjusted the manuscript accordingly and responses to all comments have been provided below (in red font).

Comments and Suggestions for Authors

Hemoglobin and its relationship with fatigue in Long-COVID 2 patients three to six months after SARS-CoV-2 infection

The authors have responded to the comments on statistical inconsistencies, and a number of other issues; I thank the authors for their substantial efforts. With the manuscript on a statistically more sound footing, I believe it is close to being suitable for publication. There are, however, a number of other issues which could helpfully be improved, mainly related to Discussion.

(1) References 1 and 2 may not be the most up-to-date citations - the authors state that four years after the WHO declared the pandemic "infections from SARS-CoV-2 and their long-term consequences remain a global challenge (1, 2)" but the cited references are from 2020 and 2021. It might be better to use current literature references. I suggest PMID: 36639608 and PMID: 36620963 as more recent examples, that are more focused on the challenges posed by your specific research. --> The references have been updated using the suggested articles.

(2) Line 117, "Finding biomarkers that can assist these patients receive optimized care is crucial, and since hemoglobin can be easily to assess, is it a feature that correcting that could be a possible treatable trait." would be better written as "Finding biomarkers that can help to direct optimized care to patients most at risk is crucial, and since hemoglobin is easily assessed by assay and has been associated with long-COVID, we believe that it is a biomarker worthy of further investigation" --> The suggested sentence has now been used in the article.

(3) Line 162, missing full stop --> This typo has been corrected.

(4) Line 205, I'm not sure I follow the logic. A better reason not to use Bonferroni would be that false-discovery correction when testing small numbers of variables which are themselves correlated can be too conservative, as the effective number of tests undertaken (hemoglobin versus inflammation) is not the same as the actual number of tests undertaken (hemoglobin versus x different measures of inflammation, which are clearly not x independent tests). For your purposes, it is probably sufficient to state that FDR was not used as it can be overly conservative especially when validating a smaller number of previously described features or in cases where the effective number of tests is smaller (due to redundancy in similar features measuring the same latent variables) and referencing PMID: 16077740 and PMID: 21451529. --> The justification regarding no necessity for FDR has been adjusted based on this point and suggested references, see lines xxx-xxx of the marked version of the manuscript: “A false discovery rate (FDR) was not used as it can be overly conservative especially when validating a smaller number of previously described features or in cases where the effective number of tests is smaller (due to redundancy in similar features measuring the same latent variables) (36, 37).”

(5) Figure labelling, both figures are currently captioned as fig.2 --> This inconsistency has been corrected.

(6) The boxplot figure would be better presented as two boxplots next to each other (with larger text to keep them readable), the current boxplots take up a lot of page space without conveying any extra information than if they were next to each other --> In the revised version these suggested changes have been applied. 

(7) Lines 431 onwards referencing Hanson et al from 2024 (PMID: 38429458) presents a very interesting point, which I think the authors should expand upon, whereby Hanson et al found a link between inflammation and Hb but your analysis did not validate this link. COVID-19 is an illness which in its short life has already seen many variants and changes in presentation, which in turn increases the risk that correlations or biomarkers seen in one dataset may not be reproducible in other cohorts, as seen in PMID: 37762673. This emphasises the need for validation work to test the robustness of biomarker and pathway analyses, which you have done here. --> This has been now added to the manuscript as following: “The fact that we could not validate the same link between inflammation and Hb that was reported by Hanson et al. emphasizes the need to further test the robustness of biomarker analyses (52). Furthermore, a meta-analysis reviewing the studies evaluating the COVID-19 biomarkers revealed a number of findings, including different results per population, which highlights the significance of validation studies (54). Especially, since long COVID is a relatively young disease correlations with biomarkers in one population should be validated in other populations.”

(8) There may also be some value in referencing post-influenza conditions, and that this is a limitation of the study (are these findings specific to COVID, or indicative of post-viral syndromes more broadly). PMID: 38104583 may be helpful in this context. --> This very interesting suggestion has been added to the limitation part of the discussion (lines 398 – 405 of the marked version of the manuscript): “First, while SARS-CoV-2 infection is the exclusive focus of this investigation, it has been revealed that other viral infections, such as seasonal influenza, may have long-term effects similar to those following SARS-CoV-2 infection (59). Thus, it is uncertain whether our findings are specific to this particular viral infectious disease, or indicative of post-viral syndromes more broadly.”

(9) In my view, the Conclusions section could be slightly expanded to mention the importance of validation studies to test whether initial findings are reproducible and robust across different cohorts. --> This point about validation studies has been added to the conclusion, see lines 447 – 449 of the marked version of the manuscript (also with bold Italic font below):

“In our study of Long-COVID patients, we observed that those with fatigue tended to have lower hemoglobin levels than those without. These findings suggest a role of hemoglobin in Long-COVID fatigue. To determine the causes of low hemoglobin levels in Long-COVID patients and in turn, the best course of action for treatment, further studies on the mechanisms influencing hemoglobin levels in Long-COVID are necessary. Furthermore, it is important to perform validation studies to examine whether our findings are reproducible and robust across different cohorts.

Comments on the Quality of English Language

No major comments.

Submission Date

11 February 2024

Date of this review

05 Apr 2024 14:10:48

Round 3

Reviewer 1 Report

Comments and Suggestions for Authors

The authors have comprehensively responded to my questions and comments, and I thank them for their efforts. In my view, the manuscript is suitable for publication and adds to the COVID-19 knowledge base.

Author Response

Thank you!